# Progression of HTLV-1 Associated Myelopathy/Tropical Spastic Paraparesis after Pregnancy: A Case Series and Review of the Literature

**DOI:** 10.3390/pathogens13090731

**Published:** 2024-08-28

**Authors:** Frederique A. Jacquerioz, Mauricio La Rosa, Elsa González-Lagos, Carolina Alvarez, Martin Tipismana, Karen Luhmann, Eduardo Gotuzzo

**Affiliations:** 1Department of Tropical Medicine, Tulane School of Public Health and Tropical Medicine, New Orleans, LA 70112, USA; fjacque@tulane.edu; 2Institute of Tropical Medicine Alexander von Humboldt, Universidad Peruana Cayetano Heredia, Lima 15102, Peru; elsa.gonzalez@upch.pe (E.G.-L.); carolina.alvarez@upch.pe (C.A.); martin.tipismana@upch.pe (M.T.); karen.luhmann@gmail.com (K.L.); 3Department of Obstetrics and Gynecology, Pennsylvania Hospital, Philadelphia, PA 19107, USA; malarosa@utmb.edu; 4Department of Neurology, Hospital Cayetano Heredia, Lima 15102, Peru

**Keywords:** human T lymphotropic virus 1, HTLV-1, pregnancy, postpartum, HTLV-1 associated myelopathy/tropical spastic paraparesis

## Abstract

HTLV-1-associated Myelopathy/Tropical Spastic Paraparesis (HAM/TSP) is a progressive non-remitting and incapacitating disease more frequently seen in women and with a patchy worldwide distribution. HAM/TSP develops in a small percentage of HTLV-1-infected individuals during their lifetime and etiologic factors for disease progression are still unclear. This study aims to describe the first case series of the progression of HAM/TSP in relation to pregnancy. Between January and March of 2012, we reviewed medical charts of women with HAM/TSP currently enrolled in the HTLV-1 cohort at the institute of tropical medicine of Cayetano Heredia University. Inclusion criteria included having a diagnosis of HAM/TSP according to the WHO guidelines and self-reported initial symptoms of HAM/TSP during pregnancy or within six months of delivery. Fifteen women reported having had symptoms compatible with HAM/TSP within four months of delivery. Among them, ten women had no symptoms before pregnancy and reported gait impairment after delivery. Five women with mild gait impairment before pregnancy noticed a worsening of their symptoms after delivery. Symptoms worsened after successive pregnancies. Recent studies have shown that HTLV-1 infection induces a strong T cell-mediated response and that the quality of this response plays a role in HAM/TSP pathogenesis. The relative immunosuppression during pregnancy, including blunting of the T-cell response, might allowed in certain women enhanced replication of HTLV-1 and disease progression in the postpartum. This is the first study looking specifically at HAM/TSP and pregnancy and the number of cases is remarkable. Further prospective studies of HTLV-1-infected women assessing immune markers during pregnancy are warranted. Breastfeeding was the main route of transmission. Strategies to prevent vertical transmission need to be evaluated in HTLV-1 endemic countries of Latin America.

## 1. Introduction

Human T lymphotropic virus 1 (HTLV-1)associated Myelopathy/Tropical Spastic Paraparesis (HAM/TSP) is a slowly progressive inflammatory disease characterized by spastic paraparesis predominantly of the lower extremities, sphincter dysfunction, and mild sensory disorder [1]. Although an uncommon disease at a global level, it disproportionally affects families and communities in the endemic foci of Japan, the Caribbean, sub-Saharan Africa, Melanesia, and South America. People acquire HTLV-1 infection through breastfeeding (infancy), sexual intercourse, blood transfusion, or organ transplantation. Among HTLV-1-infected individuals, the lifetime risk of developing HAM/TSP is 0.25 to 4% [2]. The risk seems to be even greater in the Caribbean and South America [2,3] and is three times higher in women [4,5]. Socio-economic and psychological impacts of HAM/TSP are dramatic as the disease progresses irrevocably to severe disability in the span of years and to date, there is no effective treatment [6]. However, the actual prevalence and impact of HTLV-1 in pregnant women remain largely unknown, highlighting the importance of new studies and more research on this topic. The impact is not only on the affected individuals but also on their entire family and mothers with small children suffer most of the disease burden. Host immunity plays an important role in disease pathogenesis [1,2,7,8]. However, etiologic factors of disease progression are still unknown. Pregnancy induces significant immunological changes, which could potentially influence the progression of HAM/TSP in the immediate postpartum period [9]. Since 1989, an HTLV-1 cohort study has been carried out at the Instituto de Medicina Tropical Alexander von Humboldt (IMTAvH) in Lima, Peru. The observation of a few cases of onset of HAM/TSP in the immediate postpartum period raised the hypothesis of pregnancy as a trigger for disease progression. We report on the demographic and clinical characteristics of patients with HAM/TSP after pregnancy.

## 2. Patients and Methods

Ethics Statement: This study complied with the principles outlined by the Declaration of Helsinki and was approved by the Institutional Ethics Committee of Cayetano Heredia University and the Human Research Ethics Committee of Tulane University. Informed consents were signed by all participants.

Setting: The study was conducted at the IMTAvH at the National Hospital Cayetano Heredia in Lima, Peru. Since 1989, patients with HTLV-1 infections and associated diseases have been referred to IMTAvH from major public hospitals and private health facilities in Lima and its surrounding areas and enrolled in the HTLV-1 cohort study.

Procedures: Between January and March of 2012, we reviewed the medical charts of women with a diagnosis of HAM/TSP who were currently enrolled in the cohort study. Inclusion criteria were women 15 years of age or older with an HTLV-1 infection diagnosed by enzyme-linked immunosorbent assays and confirmed by Western blot and/or pro-viral load (PVL) measured by real-time PCR, and a diagnosis of HAM/TSP who self-reported initial symptoms of HAM/TSP during pregnancy or within 6 months of delivery. The diagnosis of HAM/TSP was based on the WHO guidelines and the revised definition by Castro-Costa [10,11]. Cases identified on medical charts were contacted and those alive and agreeing to participate were interviewed and examined by a neurologist to determine their current stage of HAM/TSP, and blood samples were drawn for PVL quantification. PVL was measured as reported previously [12] and expressed as the number of HTLV-1 copies per 10^4^ peripheral blood mononuclear cells (PBMC). We collected information on demographics, risk factors for HTLV-1 infection, and clinical characteristics of HAM/TSP. Ethnic background was considered as Quechua if the participants defined themselves as such and/or they spoke the Quechua language. We considered breastfeeding as a likely route of transmission when the mother, father, and/or siblings were found HTLV-1 positive. Sexual transmission was considered when the husband but not the parents or siblings, was HTLV-1 positive. Transmission by blood transfusion was considered plausible when participants had received transfusions before 1998 when blood products started to be tested for HTLV-1 in Peru. Univariate analyses were performed using SPSS (PASW Statistics 18). When the time from delivery to onset or worsening of symptoms was less than a week, 0.5 weeks was used for calculations.

## 3. Results

Of the 1439 HTLV-1-positive women enrolled in the cohort study in March 2012, 333 had HAM/TSP (prevalent cases) and at least one pregnancy. From medical chart reviews, we identified fifteen women who had self-reported onset or worsening of symptoms of HAM/TSP in the months following pregnancy. We were able to contact twelve women and all agreed to be interviewed. Among the remaining three, two were lost to follow-up, and one died in 2008.

Demographic data and risk factors: Among the fifteen, nine women (60%) were Quechua. All women reported being breastfed, fourteen (93%) for at least six months, and eleven (73%) for more than one year. Six women (40%) were likely infected through breastfeeding. Three women received blood transfusions before 1998. One of them is a confirmed case of transmission by contaminated blood products. She received a transfusion while breastfeeding her one-year-old daughter and consequently infected her. For the other two women, transmission through this route is plausible but not confirmed. Sexual transmission was possible in three cases; for two, transmission through breastfeeding or transfusion was also possible. Eight women (53%) had one or more children with HTLV-1 infection. Eleven women (73%) had one or more family members with HTLV-1 infection. All women were negative for HTLV-2 and HIV. The median time between the onset of symptoms and diagnosis of HAM/TSP (serologic tests and clinical examination) was 6 years (range: 0–42) (Table A1 and Table A3).

Clinical and obstetric data: Of the fifteen women, ten were classified as having onset of HAM/TSP after delivery (cases #1 to #10). The mean age was 29.5 years (SD: 6.8) and the median time between delivery and onset of symptoms was 2 weeks (range: 0.5–16). Difficulty walking was the symptom most frequently reported on medical charts. In one woman, transitory cramps and weakness in the right leg during the first trimester were also reported. The remaining five women were classified as having worsening HAM/TSP after delivery (cases #11 to #15). They had a mean age of 26.2 years (SD: 7.6) when they first presented symptoms and 33.2 years (SD: 5.7) at delivery. The median time between delivery and worsening of symptoms was 1 week (range: 1–8). Difficulty walking was present before pregnancy in these women but a clear worsening of the symptoms was noted after delivery. Only two of the five had a confirmed diagnosis of HAM/TSP before pregnancy. One woman was not able to walk again after delivery and had to use a wheelchair (Table A1, Table A2 and Table A3).

During interviews, the main self-reported symptom after pregnancy was profound weakness (92%) that impeded their ability to walk normally. Stumbling (67%) (i.e., instability while walking, falls, and/or fear of falling) and numbness (42%) were also commonly reported. The symptomatology predominated in lower limbs but one patient also reported difficulties carrying her newborn in her arms. Three women (25%) also reported sphincter dysfunction. All twelve women reported that the symptoms affected their daily life activities and nine (75%) stopped working as a result of the disease. The findings of the 2012 neurologic examination are summarized in Table A3. The median duration of the disease was 16.5 years (range: 10–40). Five women were in a wheelchair after a median of 26 years (range: 9–36). A majority were receiving a muscle relaxant and three were treated with a combination of an antiretroviral, muscle relaxant, and prednisone. The score of the Kurtzke Expanded Disability Status Scale (EDSS) [13] used to assess disability in HAM/TSP patients, was 6 on a scale of 0 to 10 (range 4.5–8) [13]. Scores were not available for times of onset/worsening to assess the progression of the disease or to corroborate the above self-reported symptoms. Most women entered the cohort years after the onset/worsening of symptoms (Table A3).

## 4. Discussion

We describe fifteen HTLV-1-infected women with a progression of HAM/TSP following pregnancy. To date, the potential effect of pregnancy on HTLV-1 infection has been poorly investigated and published [14]. In 1994, Mizokami et al. [14] reported on a case of a Japanese woman with HTLV-1 infection who presented gait disturbance with progressive stiffness and weakness in her right leg four months after delivering her first child. The symptoms improved during her second pregnancy but were exacerbated five months after delivering her second child. She was then diagnosed with HAM/TSP and thyroiditis. The reason for very few reported cases might be explained by various epidemiological factors. Isolated cases might be difficult to recognize and interpret in clinical practice. Initial symptoms might be attributed to the pregnancy itself or to normal postpartum recovery. A puerperal woman in charge of a newborn might first minimize symptoms or deny them for fear of being sick [15]. The disease often affects people from low socio-economic backgrounds who, with a lack of financial resources to pay for healthcare, might delay seeking care [6,15]. More importantly, in many endemic countries, a majority of health professionals are unaware of HTLV-1 infection and its associated diseases [6,15,16]. By the time a diagnosis is confirmed, months or years have passed and by then the temporal connection between initial symptoms and pregnancy is likely to be overlooked. The recognition of our cases was made possible by an experienced HTLV-1 clinical and research team and the existence of an ongoing HTLV-1 cohort that allowed retrospective and specific evaluation of cases.

Gait impairment was the leading complaint and the main reason for seeking medical advice in our series of patients. We cannot be certain that symptoms such as weakness had not been present during pregnancy and either not recalled or thought to be due to the pregnancy itself. Similarly, less specific symptoms of HAM/TSP such as urinary incontinence or constipation could also have been present for some time but not attributed to the disease. However, after pregnancy gait impairment was clearly identified and reported by the women as unusual and an in-person interview confirmed this observation. This onset or worsening of HAM/TSP postpartum suggests that pregnancy could have elicited disease progression in these women, in particular the motor component.

The Japanese woman reported on by Mizokami [14] presented an onset of HAM/TSP after her first pregnancy and exacerbation following her second pregnancy. This was also true for the four of our patients who had another pregnancy and reinforces the hypothesis that pregnancy may play a role in HAM/TSP progression. A plausible hypothesis might be related to the role of cell-mediated immunity in both pregnancy and HAM/TSP [9,17]. Pregnancy induces immune tolerance to the fetus through a complex modulation of the immune system, which includes a shift of effector T-helper (T_H_) cells from T_H_ 1 (cell-mediated) to T_H_ 2 (antibody-mediated) dominance, expansion of FoxP3 regulatory T cells (Tregs), and down-regulation of T_H_17 cells [9,18]. All of these changes contribute to suppressing an excessive T cell-mediated immune response to fetal alloantigen and impaired function of these cells leads to inflammation and pregnancy-associated complications [9,19]. In contrast, it is now accepted that HTLV-1 induces strong cytotoxic T cell (CTL) immune response [20] and recent studies have shown that the quality of this response, partly defined by genetic factors, is important in determining disease progression [21,22,23,24,25]. In asymptomatic carriers, CTL response mounted against HTLV-1 seems to be efficient in killing virally infected T cells, mostly Tregs expressing FoxP3, and maintaining low proviral loads [23,24,25]. In HAM/TSP patients, an exaggerated inflammatory response characterized by higher PVLs, overproduction of pro-inflammatory cytokines such as interferon γ, and accumulation of anti-HTLV-1 CTLs in the cerebrospinal fluid is observed. In these patients, the underlying mechanism is thought to be due to deregulation in function and frequency of CTLs and subsets of Tregs expressing FoxP3 [17,21,24,26,27,28,29]. Thus, one can postulate that the relative but critical immunosuppression during normal pregnancy, including blunting of the CTL response, alters the Th1/Th2 immune response equilibrium of HTLV-1 infected women allowing enhanced replication of HTLV-1, increased inflammation, and disease progression postpartum.

A comparable pattern of disease progression or relapse in postpartum is observed in other inflammatory and T cell-mediated dominant diseases such as multiple sclerosis (MS), and rheumatic arthritis (RA) [30,31]. During pregnancy, MS relapses less frequently and is often managed without medication. In post-partum, MS flares are more severe and occur in almost 30% of women within 3 months of delivery [32,33,34]. Similarly, RA is five times more likely to develop after delivery than at any other time [32]. The exact mechanism of this phenomenon is not yet completely elucidated.

Studies have investigated the use of steroids in the management of HAM/TSP in HTLV-1 patients. It aims to alleviate symptoms and reduce inflammation in affected individuals. While some studies have reported short-term benefits, such as a reduction in inflammation and improvement in motor function, the long-term efficacy of steroid therapy remains uncertain. The prolonged use requires screening of other infectious diseases, otherwise it could carry the risk of adverse effects, such as infectious diseases or metabolic disturbances [35,36]. Given the hypothesis that immune suppression during pregnancy and its subsequent release postpartum may trigger or worsen HAM/TSP, further research is needed to explore the efficacy and safety of steroid administration during the postpartum period for patients with HAM/TSP. This research could help clarify whether steroids might be beneficial in managing postpartum exacerbations of HAM/TSP in the context of HTLV-1 infection.

In HTLV-1 infection, the effect of pregnancy on HTLV-1 infection and HAM/TSP might vary by geographic regions, host genetic determinants, mode of HTLV-1 acquisition, or virus subtypes and a combination of various factors may need to be present to elicit disease progression. There are no published observational studies of HTLV-1-infected pregnant women looking specifically at this issue. In 2003, Ando et al. [37]. described two cases of HAM/TSP diagnosed during pregnancy, one during the first trimester and the other at 22 weeks of gestation, suggesting opposing findings. However, it is unclear whether the symptoms were present before pregnancy but noticed during pregnancy, and their characteristics after delivery were not reported. In a case-control study conducted in Gabon in the early 1990s assessing the effect of HTLV-1 infections on maternal and fetal outcomes, the authors mentioned briefly that none of the 45 HTLV-1 positive cases presented HAM/TSP or other clinical manifestations of HTLV-1 infection during pregnancy, after delivery, or at one-year follow-up [38]. In 2021, Mori et al. [39] in Japan documented a case report of HTLV-1 infection concurrent with early-onset HAM/TSP during pregnancy. Their report indicated the absence of a complication pertaining to the gait impairment and/or neurological symptoms or worsening of symptoms. However, it is crucial to note that the case they described was characterized by severe HAM/TSP symptoms, including severe gait impairment (use of a wheelchair), prior to the onset of pregnancy. In Latin America, this phenomenon needs to be further tested through careful observational studies of asymptomatic HTLV-1 infected women during pregnancy and post-partum looking at genetic determinants, immune response, virus factors, and sexual hormones as potential factors in disease progression. Such studies will likely shed light on the role of pregnancy in HAM/TSP and bring opportunities for disease prevention and new targets for drug development. Additionally, it is necessary to study and identify that immunological changes during pregnancy and postpartum may have an effect on the development of complications associated with HTLV-1 [16,40].

In most Latin American countries including Peru, HTLV-1 screening of pregnant women is not done routinely and HTLV-1 awareness programs are nonexistent outside of a few specialized medical centers. In our series, six women (40%) were probably infected by their mothers during breastfeeding and at least eight (53%) transmitted HTLV-1 to their infants. These numbers might be higher as some relatives and children were not tested for HTLV-1 infection. The rate of transmission clearly increases with the length of breastfeeding from 4% when children are breastfed less than 6 months to almost 40% with prolonged breastfeeding (more than 1 year) [41,42,43,44,45]. In Peru, breastfeeding for up to 2 years is a common practice and is highly promoted as a strategy to reduce child morbidity and mortality, especially among women from lower socio-economic status. Therefore, interventions to screen pregnant women and prevent HTLV-1 vertical transmission should carefully balance the vital benefit of breastfeeding practices with the risk of transmission. In this perspective, counseling of HTLV-1 pregnant women and personalized interventions depending on socio-economic resources, familial support, and availability of national programs are crucial. In addition, if pregnancy does indeed play a role in disease progression, counseling of HTLV-1 infected women should be broadened to inform them about these risks.

Despite limitations related to the study design, the number of cases is high enough to be intriguing. We were only able to identify women who spontaneously reported symptoms in relation to pregnancy during previous medical visits; as a result, a comparison with other women in the cohort could not be performed. The lack of a control group limits our ability to directly assess the impact of pregnancy on the onset or worsening of HAM/TSP. Additionally, there is a lack of comprehensive literature specifically addressing the impact of pregnancy on HAM/TSP. The in-person interviews might have introduced recall bias in patients who presented the event of interest many years ago. For instance, they might have better recalled motor impairment than urinary or bowel disorder. We minimized this bias by comparing data from interviews to data from medical charts whenever possible.

## 5. Conclusions

There is an urgent need to better understand the disease pathogenesis of HAM/TSP and the triggers of disease progression to prevent irreversible neurologic damage. Also, the role of proviral load before, during, and after pregnancy should be studied to find correlations. This case series provides new insights into the possible role of gestational and postpartum immune changes on HAM/TSP progression that should be further evaluated. It has potential implications for counseling HTLV-1-infected women regarding pregnancy and in the development of novel therapeutic and preventive approaches. In addition, strategies to prevent vertical transmission and to counsel alternative feeding practices among HTLV-1-infected mothers should be developed and evaluated in endemic regions of Latin America, working towards stemming the tide of HTLV-1 transmission and reducing the burden of associated complications.

## Data Availability

Data is not available publicly due to privacy and ethical restrictions.

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
