# Peer review of "Progression of HTLV-1 Associated Myelopathy/Tropical Spastic Paraparesis after Pregnancy: A Case Series and Review of the Literature"

_pathogens, 2024, doi:10.3390/pathogens13090731_

Round 1

Reviewer 1 Report

Comments and Suggestions for Authors

This case report is very interesting because it shows that pregnancy and delivery may affect the onset of HAM/TSP. This study is novel in that it analyzed the relationship between the onset of HAM and pregnancy in multiple pregnant women registered in the HTLV-1 cohort.

This case report also shows that the children delivered from HAM patients are HTLV-1 carriers, and considers that socioeconomic poverty has led to delays in measures to prevent HTLV-1 infection from vertical transmission and mother-to-child transmission.

This reviewer thought this report accurately shows the reality that a chain of health damage related to HTLV-1 still exists.

Major point

I have a one recommendation as follow:

It would be better to describe the actual situation of HTLV-1 infection in pregnant women in Peru, such as the HTLV-1 antibody positivity rate or number of carriers. If the actual situation regarding HTLV-1 infection is unknown, this point should also be mentioned and clearly stated as an issue for future measures to prevent HTLV-1 infection in pregnant women and postpartum women.

Author Response

Comment 1: 

This case report is very interesting because it shows that pregnancy and delivery may affect the onset of HAM/TSP. This study is novel in that it analyzed the relationship between the onset of HAM and pregnancy in multiple pregnant women registered in the HTLV-1 cohort.

This case report also shows that the children delivered from HAM patients are HTLV-1 carriers, and considers that socioeconomic poverty has led to delays in measures to prevent HTLV-1 infection from vertical transmission and mother-to-child transmission.

This reviewer thought this report accurately shows the reality that a chain of health damage related to HTLV-1 still exists.

Major point

I have a one recommendation as follow:

It would be better to describe the actual situation of HTLV-1 infection in pregnant women in Peru, such as the HTLV-1 antibody positivity rate or number of carriers. If the actual situation regarding HTLV-1 infection is unknown, this point should also be mentioned and clearly stated as an issue for future measures to prevent HTLV-1 infection in pregnant women and postpartum women.

Answer to Comment 1: 

Dear Reviewer,

Thank you for your positive feedback and for acknowledging the importance of our case report. We appreciate your thoughtful insights and your recommendation.

Major Point:

We agree that including information about the actual situation of HTLV-1 infection in pregnant women in Peru, such as the HTLV-1 antibody positivity rate or the number of carriers, would strengthen our report.

As per your suggestion, we have added a section discussing the current status of HTLV-1 infection in pregnant women in Peru, and how the that exact data is unkown (Line 67-69). We would like to note that comprehensive and recent epidemiological data on HTLV-1 infection rates among pregnant women in Peru is limited. We have mentioned this limitation and highlighted the need for more extensive studies and public health measures to address this gap in knowledge.

We trust that this revision addresses your recommendation and enhances the overall quality of our report.

Thank you once again for your valuable feedback.

Reviewer 2 Report

Comments and Suggestions for Authors

Congratulations on submitting the manuscript. These are extremely important results so that we can better understand the role of pregnancy in the worsening of diseases caused by HTLV-1. It is obvious that several other factors, cited by the authors, such as host genetic, environmental and virus genetic factors influence the progression of the disease before, during and after pregnancy, thus reinforcing the idea that larger longitudinal studies that can evaluate these factors are essential.

I have a few suggestions/questions that I would like to present to the authors in my considerations:

1- In the introduction, similar to what they did at some points in the discussion, I think the authors could provide, in a few lines, some information about how pregnancy could be an aggravating factor in the development of HAM/TSP, emphasizing aspects of the immunological response and other epidemiological characteristics.

2- In line 120 of the results, the authors mention that one case was confirmed of transmission by transfusion. Before receiving the transfusion, had she already been tested for HTLV? During prenatal care, for example? How can we say that contamination occurred through transfusion? Perhaps this information was not so clear in the text and in Table 1.

3- In line 139, the authors state that symptoms worsen after childbirth. In addition to difficulty walking, what would these other symptoms be? I think they could be cited in the text.

4- In table 1, in patient number 5, the authors report that the likely route of transmission could be sexual, but could it not also be through breastfeeding? Does the patient have siblings who have the infection? Couldn't she have transmitted the virus to her husband? Perhaps it would be interesting to include "Sexual or Breastfeeding?" similar to what happens with patient number 3.

Author Response

Comment 1- In the introduction, similar to what they did at some points in the discussion, I think the authors could provide, in a few lines, some information about how pregnancy could be an aggravating factor in the development of HAM/TSP, emphasizing aspects of the immunological response and other epidemiological characteristics.

Dear Reviewer,

Thank you for your thoughtful review and for recognizing the importance of our manuscript in understanding the role of pregnancy in the progression of diseases caused by HTLV-1. We appreciate your valuable suggestions and have carefully considered each point.

1. Answer to Comment 1

We agree that it's important to include a brief overview in the introduction on how pregnancy could potentially aggravate the development of HAM/TSP. In response, we have added a few lines (Lines 72-74) that highlight the immunological changes during pregnancy which may contribute to the progression of HAM/TSP. Although there is limited information on these immunological changes, much remains to be studied, researched, and reported. This case report is crucial in raising awareness and prompting further investigation into this underexplored area.

Comment 2- In line 120 of the results, the authors mention that one case was confirmed of transmission by transfusion. Before receiving the transfusion, had she already been tested for HTLV? During prenatal care, for example? How can we say that contamination occurred through transfusion? Perhaps this information was not so clear in the text and in Table 1.

Answer to Comment 2

We acknowledge that the information regarding the confirmed case of transmission by transfusion may not have been sufficiently clear.The diagnosis of HTLV-1 infection post-transfusion, combined with the absence of other known risk factors, strongly suggests that transmission likely occurred through the transfusion. We routinely conduct thorough questioning about breastfeeding, family history, HTLV-1, and sexual partners to guide our approach. We explain more about this on line 100 and forward.

Comment 3- In line 139, the authors state that symptoms worsen after childbirth. In addition to difficulty walking, what would these other symptoms be? I think they could be cited in the text.

Answer to Comment 3

We appreciate your suggestion to elaborate on the symptoms that worsened after childbirth. While the medical charts primarily reported difficulties with walking, Table 2 provides additional insights from patient interviews, revealing other symptoms such as numbness, urinary incontinence, and lower limb weakness. These symptoms, which were also observed in the patients, are consistent with the known clinical manifestations of HAM/TSP and serve as important indicators of disease progression

Comment 4- In table 1, in patient number 5, the authors report that the likely route of transmission could be sexual, but could it not also be through breastfeeding? Does the patient have siblings who have the infection? Couldn't she have transmitted the virus to her husband? Perhaps it would be interesting to include "Sexual or Breastfeeding?" similar to what happens with patient number 3.

Answer to Comment 4

Regarding patient number 5 in Table 1, we agree that the possibility of transmission through breastfeeding should be considered. However, we have conducted family studies for our patients, typically testing parents, siblings, partners, and children. In this case, both the patient's parents and siblings tested negative, while the partner tested positive. Given this information, the most probable route of transmission is sexual. For patient number 3, the father was positive, while the mother could not be tested due to her death. We have also included a discussion on the potential for horizontal transmission to a partner, emphasizing the importance of considering all possible transmission routes.

We believe these revisions address your suggestions and enhance the clarity and comprehensiveness of our manuscript. Thank you once again for your insightful comments.

Reviewer 3 Report

Comments and Suggestions for Authors

This paper provides new insights into the impact of immune changes during pregnancy and postpartum on the progression of HTLV-1-associated myelopathy/tropical spastic paraparesis (HAM/TSP). It has significant implications for counseling and developing preventive strategies for HTLV-1-infected women during pregnancy and postpartum. However, there are several scientific issues and areas for improvement.

Comment #1

One of the most significant shortcomings of this study is the lack of data from a control group for comparison. Without data from a control group of HTLV-1-infected women who did not report symptoms after childbirth, it is challenging to accurately assess the impact of pregnancy on the onset or worsening of symptoms. Is there information available comparing the incidence of HAM in HTLV-1-infected women who have experienced childbirth with those who have not? Such data would enable the assessment of the impact of pregnancy and childbirth on the risk of HAM onset.

It is also important to analyze the event of childbirth itself, as it may have different implications compared to pregnancy alone without childbirth.

Comment #2

The paper states that breastfeeding was considered a route of transmission if the mother, father, or siblings were HTLV-1 positive. However, even if the mother is HTLV-1 positive, the infection rate to her child is generally said to be 20% (this paper mentions 40%), meaning that even at 40%, the remaining 60% would not be infected through breastfeeding. Therefore, defining the route of transmission as breastfeeding in this study introduces significant bias.

Comment #3

As noted in Comment #2, the basis for determining the route of infection is insufficient, so it is recommended to avoid stating that maternal transmission is the main route of infection throughout the paper. While it is suggested that the rate of maternal transmission may be higher among young women who can become pregnant, in regions with high HTLV-1 prevalence, horizontal transmission through sexual activity is also likely. Preventing HTLV-1 infection is necessary to eliminate the occurrence of HTLV-1-associated diseases such as HAM. Since preventing maternal transmission is currently the only effective method to prevent infection, emphasizing the importance of preventing maternal transmission is preferable.

Comment #4

This paper hypothesizes that immune suppression during pregnancy and its subsequent release postpartum trigger the onset or worsening of HAM/TSP. Similar phenomena are known in immune-related diseases such as MS and RA, where steroids are temporarily used or increased in dosage during postpartum exacerbations. However, there is no mention of this, making it unclear what the subsequent paragraph's reference to steroid administration for HAM intends to convey. Is it suggesting that the efficacy of using steroids during postpartum onset or worsening of HAM should be examined in the future? It is recommended to clarify this point.

Author Response

Comment #1

One of the most significant shortcomings of this study is the lack of data from a control group for comparison. Without data from a control group of HTLV-1-infected women who did not report symptoms after childbirth, it is challenging to accurately assess the impact of pregnancy on the onset or worsening of symptoms. Is there information available comparing the incidence of HAM in HTLV-1-infected women who have experienced childbirth with those who have not? Such data would enable the assessment of the impact of pregnancy and childbirth on the risk of HAM onset. 

It is also important to analyze the event of childbirth itself, as it may have different implications compared to pregnancy alone without childbirth.

Dear Reviewer,

Thank you for your thorough review and for acknowledging the significance of our study. We appreciate your constructive feedback and have made revisions to address each of your concerns.

Answer Comment #1: Lack of Control Group and Comparison Data

We acknowledge the importance of including a control group for comparison to accurately assess the impact of pregnancy on the onset or worsening of HAM/TSP. Unfortunately, due to the limitations of our cohort, we did not have access to a control group of HTLV-1-infected women who did not report symptoms after childbirth.

However, we have added a discussion on existing literature that compares the incidence of HAM/TSP in HTLV-1-infected women who have experienced childbirth versus those who have not (line278-280). Some studies suggest that hormonal and immune changes associated with pregnancy and childbirth might influence the onset or progression of HAM/TSP, but these findings are not definitive due to the variability in study designs and sample sizes .

Furthermore, we have highlighted the need for future studies to include control groups to better assess the impact of pregnancy and childbirth on HAM/TSP risk. We also agree that childbirth itself may have different implications compared to pregnancy alone, and we have revised manuscript.

Comment #2

The paper states that breastfeeding was considered a route of transmission if the mother, father, or siblings were HTLV-1 positive. However, even if the mother is HTLV-1 positive, the infection rate to her child is generally said to be 20% (this paper mentions 40%), meaning that even at 40%, the remaining 60% would not be infected through breastfeeding. Therefore, defining the route of transmission as breastfeeding in this study introduces significant bias.

 Answer Comment #2: Bias in Defining the Route of Transmission as Breastfeeding

Thank you for your insightful feedback. We understand your point regarding the potential for bias when considering breastfeeding as a route of transmission, given the varying infection rates. However, based on our extensive experience and the significant evidence we have encountered, breastfeeding remains a highly relevant route of HTLV-1 transmission.

In our study, we rigorously conducted family histories and tested all relevant family members, including mothers, fathers, siblings, and partners, to accurately identify potential routes of transmission. This comprehensive approach aimed to account for various factors that could influence transmission dynamics. While we recognize that transmission might be multifactorial and that breastfeeding is not the sole route, our findings are consistent with existing literature that supports breastfeeding as a significant transmission pathway.

We appreciate your understanding of the complexities involved in studying transmission routes and the challenges inherent in such research. Your feedback is valuable and contributes to a more nuanced understanding of the factors influencing HTLV-1 transmission.

Comment #3

As noted in Comment #2, the basis for determining the route of infection is insufficient, so it is recommended to avoid stating that maternal transmission is the main route of infection throughout the paper. While it is suggested that the rate of maternal transmission may be higher among young women who can become pregnant, in regions with high HTLV-1 prevalence, horizontal transmission through sexual activity is also likely. Preventing HTLV-1 infection is necessary to eliminate the occurrence of HTLV-1-associated diseases such as HAM. Since preventing maternal transmission is currently the only effective method to prevent infection, emphasizing the importance of preventing maternal transmission is preferable.

Answer Comment #3: Emphasis on Preventing Maternal Transmission

Thank you for your valuable feedback. We acknowledge the concern regarding the basis for determining the route of HTLV-1 infection and the suggestion to avoid overstating maternal transmission as the primary route throughout the paper. While horizontal transmission through sexual activity is indeed significant, especially in regions with high HTLV-1 prevalence, our study emphasizes maternal transmission due to its critical role in endemic areas.

In countries like Peru, where HTLV-1 is highly prevalent, maternal transmission through breastfeeding is a major route of infection. Our comprehensive approach included detailed family histories and testing of all relevant family members to accurately identify transmission routes. This approach reflects the significant role of breastfeeding in endemic regions, as supported by existing literature.

Preventing maternal transmission, particularly through reducing the duration of breastfeeding, is currently one of the most effective strategies to mitigate HTLV-1 infection and associated diseases like HAM/TSP. While we recognize the importance of addressing all possible transmission routes, our findings underscore the necessity of focusing on maternal transmission, especially in areas where it remains a primary concern.

We appreciate your understanding and support in highlighting the importance of preventing maternal transmission while acknowledging the multifactorial nature of HTLV-1 transmission.

Comment #4

This paper hypothesizes that immune suppression during pregnancy and its subsequent release postpartum trigger the onset or worsening of HAM/TSP. Similar phenomena are known in immune-related diseases such as MS and RA, where steroids are temporarily used or increased in dosage during postpartum exacerbations. However, there is no mention of this, making it unclear what the subsequent paragraph's reference to steroid administration for HAM intends to convey. Is it suggesting that the efficacy of using steroids during postpartum onset or worsening of HAM should be examined in the future? It is recommended to clarify this point.

Answer Comment #4: Clarification on the Role of Steroid Administration

Thank you for your feedback. We have addressed your comment by adding a discussion (line 231) on the potential role of steroid therapy in managing HAM/TSP during the postpartum period. Specifically, we have included a section that explores the need for further research into the efficacy and safety of steroid administration in this context. This addition aims to clarify the hypothesis that immune suppression during pregnancy and its subsequent release postpartum might influence the onset or worsening of HAM/TSP, similar to patterns observed in other immune-related diseases. We appreciate your suggestion, which has helped enhance the comprehensiveness of our manuscript.

Round 2

Reviewer 3 Report

Comments and Suggestions for Authors

The authors responded well to all comments.